



# Glacier terminus retreat, mass budget, and surface velocity measurements for the Jankar Chhu Watershed, Lahaul Himalaya, India

Suresh Das[*], Milap Chand Sharma

Centre for the Study of Regional Development (CSRD), Jawaharlal Nehru University, New Delhi 110067, India

[*]Correspondence to: suresh41_ssf@jnu.ac.in

**Abstract**

Characterization of glacier changes in the terminus, elevation, and surface velocity was worked out for the Jankar Chhu Watershed (JCW) of Lahaul Himalaya using freely available satellite remote sensing data and the limited number of field observations. We studied changes using Corona (1971), Landsat (1993–2017), Sentinel 2A (2016), the SRTM Digital Elevation Model (DEM; 2000), and the global TanDEM–X DEM (2014). Change detection for nine select glaciers was studied in detail. Our results showed that changes in glacier terminus ($-4.7 \pm 0.4$ m a$^{-1}$) between 1971 and 2016 are smaller than previously reported. An intricate pattern of mass changes across the JCW was observed, with surface lowering on an average of $-0.7 \pm 0.4$ m a$^{-1}$ which equates to a geodetic mass balance of $-0.6 \pm 0.4$ m w.e. a$^{-1}$ during 2000–14. The computed glacier surface velocities (1993–2017) reveal nearly stagnant debris-covered ablation zone but the dynamically active main trunk. Field observations/measurements also support the findings. This study provides valuable insights into the recent glacier variations, which are of critical importance to assess the future glacier dynamics on a regional scale in areas like the present one. The dataset is freely accessible at http://doi.org/10.5281/zenodo.3383233 (Das and Sharma, 2019b).

**Keywords:** terminus retreat; geodetic mass balance; surface velocity; debris-covered glacier; remote sensing; Jankar Chhu Watershed; Lahaul Himalaya

## 1 Introduction

Alpine glaciers are regarded as one of the best indicators of climate change, including its' contribution to sea-level rise (Immerzeel et al., 2010; Bolch et al., 2012). The great Asiatic mountain chains such as Hindu Kush–Karakoram–Himalaya (HKH) contains some of the enormous ice masses outside the polar regions which replenish the primary perennial river system of the south and southeast Asia (Immerzeel et al., 2010). These Himalayan glaciers have generated much debate in the last few years, mainly concerning: (a) potential consequences of glacier change on regional water availability (Immerzeel et al., 2010), and (b) understanding the dynamics of the glacier with changing climate (Bolch et al., 2012; Scherler et al., 2011b).

Similar to other regions of the world, Himalayan glaciers have been in a general state of recession since the 1850s (Mayewski and Jeschke, 1979), except for an emerging indication of stability or mass gain in the Karakoram (Hewitt,



2005; Bhambri et al., 2017). Glaciers in the central Himalaya (between Uttarakhand in the west and Bhutan in the east ) are receding at different rates (Bhambri et al., 2011; Basnett et al., 2013; Racoviteanu et al., 2015) than the western Himalayan glaciers (Schmidt and Nüsser, 2012, 2017; Chand and Sharma, 2015; Das and Sharma, 2019a) (Fig. 1a). Regional climatic differences between the monsoonal and cold–arid region of the Himalaya is attributed to

heterogeneous glacier surface area loss (Immerzeel et al., 2010).

In this context, changes in surface area, length, debris cover, ELA, elevation, and velocity may assist in ascertaining glacier status in the western Himalaya, which can be easily mapped using remote sensing techniques (Schmidt and Nüsser, 2012, 2017; Das and Sharma, 2019a; Scherler et al., 2008; Gardelle et al., 2013; Vijay and Braun, 2016). Declassified high resolution imagery of Corona and Hexagon acquired during the same period as Survey of India (SoI)

topographical maps of the 1960s and 1970s provide great potential to derive the historic glacier outlines for comparison at that point of time itself and with the contemporary glacier outlines derived from high-resolution satellite images of recent years (Schmidt and Nüsser, 2012, 2017; Racoviteanu et al., 2015; Chand and Sharma, 2015). Inaccessible mountainous setting limits in–situ monitoring of essential glacier change parameters. Earlier studies in the western Himalaya mostly used the 1960s SoI topographical maps for delineation of historic glacier boundaries and

compared them with recent field surveys or remote sensing datasets for change detection (see Table S1 for details). The reported glacier change rate varied from 8% (Miyar basin) to 30% (Bhaga basin). However, accuracy issues are likely to exist in most of the earlier measurements due to the use of topographic maps for comparison (Bhambri et al., 2011; Chand and Sharma, 2015). Some studies have been published on geodetic mass balance measurements of the glacier of Lahaul–Spiti district (Vijay and Braun, 2016; Berthier et al., 2007; Mukherjee et al., 2018). To the best of

our information, no study exists on the temporal variation in terminus, elevation, and velocity past 1971 in the JCW of Lahaul Himalaya.

The previous study by Das and Sharma (2019a) comprehensively assessed the glacier surface area change in the JCW between 1971 and 2016 based on Corona, Landsat, and Sentinel images. A total of 153 glaciers were mapped in 2016, covering an area of $185.6 \pm 3.8$ km². Glacier in the JCW deglaciated at a rate of $0.2 \pm 0.1\%$ $a^{-1}$ during studied period,

which is much lower than previously reported. The potential influence of climatic and non–climatic factors on glacier change also evaluated on a regional scale. In the present study, an attempt has been made to address these following objectives:

- i) to assess the status of glaciers by addressing the changes in terminus between 1971 and 2016;
- ii) to measure the elevation change rate and geodetic mass balance during 2000–14; and

- iii) to assess the surface velocity of glaciers between 1993 and 2017;

## 2 The Jankar Chhu Watershed

The JCW is located in the Greater Himalaya range of Lahul–Spiti district in Himachal Pradesh, bordered by Zanskar River basin of Jammu and Kashmir state in the north (Fig. 1a). The Jankar Chhu (rivulet) is a tributary of Bhaga River and that confluence at Darcha (32°40´N and 77°12´E; ~3313 m a.s.l) (Fig. 1b). Bhaga River is a tributary of mighty

Chenab which contributes to the Indus River system. The total area of the JCW is ~695 km², with an altitudinal range between ~3305 and 6309 m a.s.l.



The climate of the study area is characterized by a long winter season from mid-November to March; with a spring season that lasts until the end of May (Owen et al., 1996). This region falls under the monsoon–arid transition zone: the area is influenced by both the South Asian Monsoon in the summer season and Mid–Latitudes Westerlies in the

winters. The geomorphic system of the region is dominated by the steep and high activity glaciers, which carry large amounts of supraglacial debris (Owen et al., 1995). Mass movements, particularly talus development and debris flow fans have produced major modifications of earlier landforms. Periglacial processes have been regarded as a significant landscape forming process in this region (Owen et al., 1995).

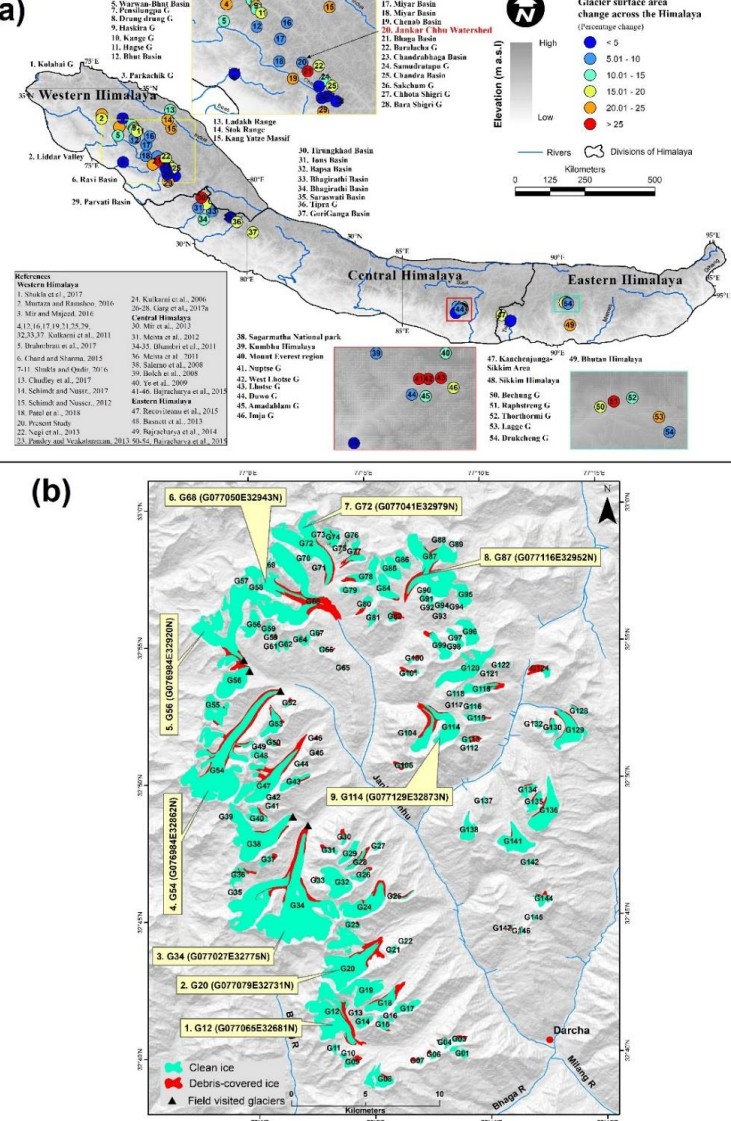




**Figure 1.** Study area. (a) Glacier surface area change (%) across the Himalayan region. (b) Spatial distribution of glaciers in the Jankar Chhu Watershed in 2016 with an individual glacier ID. The location of the JCW in the western Himalaya is shown in red color text with a black arrow in the subset (1a). For details of references in Fig. 1(a), see Supplementary Table 1. Glaciers (GLIMS ID) in callout are studied in detail.


### 3 Data sources

#### 3.1 Satellite imagery: Corona, Landsat, and Sentinel 2A

Three forward-looking Corona KH–4B images with a spatial resolution of ~1.8 m at the nadir of 27 September 1971 were used to extract the glacier length of the ~1970s (Table 1). Landsat images between 1993 and 2017 were used for

length and surface velocity measurements (Table 1). Sentinel 2A image (multispectral and orthorectified; 1 November 2016) was used to map most updated mapping. Corona KH–4B, Landsat, and Sentinel 2A images were obtained from USGS earth explorer site (http://earthexplorer.usgs.gov/) (Table 1). Also, to assist the glaciers mapping of 2016, high–resolution images available in Google Earth® software (https://www.google.com/earth/) were also used. All satellite images were rectified and co–registered to the base Sentinel image (see Das and Sharma (2019a) for details).

#### 3.2 DEMs: shuttle radar topography mission synthetic aperture radar (SRTM) and TanDEM–X

The SRTM mission was flown between 11 and 22 February 2000 and resulted with a DEM covering latitudes from 56°S to 60°N using single-pass interferometry (Farr et al., 2007). The SRTM employed two synthetic aperture radars, a C band system (5.6 cm; C-$_{band}$) and an X band system (3.1 cm; X-$_{band}$). NASA's Jet Propulsion Laboratory (JPL) was responsible for C-$_{band}$ with a swath width of 225 km and German Space Center (DLR)–Italian Space Agency (ASI)

was responsible for X-$_{band}$ with a swath width of 45 km. As a result, the X-$_{band}$ DEM has significant gaps between the strips while the C-$_{band}$ DEM is covering the area nearly without gaps. The one arc-second (~ 30 m spatial resolution) C-$_{band}$ and X-$_{band}$ SRTM DEMs were freely obtained from the USGS and the German Aerospace Center (DLR), respectively. In this study, the C-$_{band}$ SRTM DEMs were used to monitor the glacier elevation changes. The X-$_{band}$ SRTM DEM was used to evaluate the radar penetration depth of the C-$_{band}$. The C-$_{band}$ and the X-$_{band}$ DEM have the

same horizontal reference (WGS84 datum) but differ in terms of vertical reference (C-$_{band}$: WGS84 ellipsoid and X-$_{band}$: EGM96 geoid) (Farr et al., 2007).

The German TanDEM–X mission is a TerraSAR–X add–on for the generation of global high-resolution DEM with unprecedented accuracy (Rizzoli et al., 2017; Wessel et al., 2018). These twins satellite scanned the globe at least twice using X-$_{band}$ single-pass interferometer between December 2010 and early 2015. In this present study, freely

available 90 m TanDEM–X global DEM (henceforth denoted as TanDEM) to assess the glacier elevation change and mass budget over a decade. Two TanDEMs between 2011 and 2014 were acquired from the Earth Observation Center (EOC) of the German Aerospace Center (DLR) (Table 1). Since the TanDEM coverage is till January 2015, the end date for our elevation change calculation has considered being 2014.

Also, the Advanced Spaceborne Thermal Emission and Reflection Radiometer Global DEM (ASTER GDEM v2; 30m

spatial resolution) from Japan Space Systems (http://gdem.ersdac.jspacesystems.or.jp/) was used for semi-automatic delineation of drainage basin and extraction of topographic parameters (Table 1).

**Table 1.** List of all satellite and DEM datasets used in the study.

| Date of acquisition [DD-MM-YYYY] | Sensor | Resolution (m) | Path/Row/tile | Use |
|---|---|---|---|---|
| (a) satellite imagery | | | | |
| 28-09-1971 | Corona KH–4B | ~1.8 | DS1115-2282DF058 | length measurements |
| 28-09-1971 | Corona KH–4B | ~1.8 | DS1115-2282DF059 | length measurements |
| 28-09-1971 | Corona KH–4B | ~1.8 | DS1115-2282DF060 | length measurements |
| 20-10-1993 | Landsat TM | 30 | 147/37 | cross-correlation |
| 23-10-1994 | Landsat TM | 30 | 147/37 | cross-correlation |
| 02-08-2002 | Landsat ETM+ | 15 | 147/37 | cross-correlation |
| 28-08-2000 | Landsat ETM+ | 15 | 147/37 | cross-correlation |
| 15-10-2000 | Landsat ETM+ | 15 | 147/37 | length measurements |
| 30-09-2009 | Landsat TM | 30 | 147/37 | cross-correlation |
| 22-10-2011 | Landsat TM | 30 | 147/37 | cross-correlation |
| 01-11-2016 | Sentinel 2A MSI | 10 | T43SGS | length measurements |
| 19-10-2016 | Landsat OLI | 15 | 147/37 | cross-correlation |
| 06-10-2017 | Landsat OLI | 15 | 147/37 | cross-correlation |
| (b) digital elevation models | | | | |
| 11-22 February 2000 | SRTM C-$_{band}$ | 30 | N32E076 | elevation change |
| 11-22 February 2000 | SRTM X-$_{band}$ | 30 | E070N30 | radar penetration correction |
| January 2011-August 2014 | TanDEM X | 90 | N32E076 & N32E077 | elevation change |
| October, 2011 | ASTER GDEM V2 | 30 | 0N32E077 | extraction of topographic parameters |

## 4 Methods

### 4.1 Field observation and mapping

Field observations between 2015 and 2018 revealed that G34 glacier (largest glacier in the JCW) is sheltered with extensive and thick debris cover with a huge ice cliff at the terminus. A small proglacial lake exists in front of the G34 glacier. G54 glacier is characterized by a thick layer of debris on either side of the terminus, thin layer of greyish black debris in the central part of the ablation zone. Also, numerous glacier tables, subglacial tunnels, and supraglacial streams were recorded on the G54 glacier. The G56 glacier is characterized by multiple snouts and thick debris layer at the terminus. Several pro-glacier lakes were recorded in front of G56 glacier. Debris thickness along with the temperature of debris profile and underneath ice was measured for the ablation part of G34 and G68 glacier using Infrared (IR) Laser Thermometer (Fig. 2). Snout location of six glaciers was measured using handheld Global Positioning System (GPS; Garmin etrex10 with ± 5–10 m horizontal accuracy). Field measurements reveal that debris thickness on glaciers varies between ~5 and ~120 cm (Das and Sharma, 2019a). One proglacial lake has emerged in the terminus zone of G56 glacier between 2000 and 2016 (Fig. 3).

**a** 77°3'30"E  77°3'45"E  77°4'0"E

Elevation: 4678 m
Total DCT: 53 cm.
DCT: Temp (℃)
Surface: 9.0
20 cm: 4.6
30 cm: 3.2
40 cm: 1.8
Ice: -1.0

Debris thickness
measurements points
(DCT)
Ice cliffs (2016)
G68 Glacier outline

Elevation: 4712 m
Total DCT: 90 cm.
DCT: Temp (℃)
Surface: 7.5
20 cm: 6.2
40 cm: 5.8
80 cm: 1.6
Ice: -3.2

Elevation: 4676 m
Total DCT: 63 cm.
DCT: Temp (℃)
Surface: 7.6
20 cm: 7.2
40 cm: 6
Ice: -1.8

Elevation: 4705 m
Total DCT: 57 cm.
DCT: Temp (℃)
Surface: 5.2
30 cm: 4.8
40 cm: 2.8
Ice: -1.2

Elevation: 4707 m
Total DCT: 40 cm.
DCT: Temp (℃)
Surface: 9.6
5 cm: 8.8
10 cm: 4.6
20 cm: 3.4
Ice: -1.0

0   200   400
Meters

Image ©2019 Google Map

**Figure 2.** Characteristics of the ablation zone of G68 glacier (G07705032943E) in the JCW. (a) Field measurements of debris thickness measurements. The base map is a high-resolution Digi-globe image available on Google Earth. (b) The terminus of G68 glacier in 2018. (c) Temperature measurements of various debris layer and ice surface using IR Laser Thermometer. (d) Thick debris profile at the terminus zone. (d) Ice cliffs and surface morphology of ablation zone. Most probable sites for future supraglacial lakes development.

(a) Evolution of two pro-glacier lakes at the terminus zone of G56 glacier (G076984E32920N).

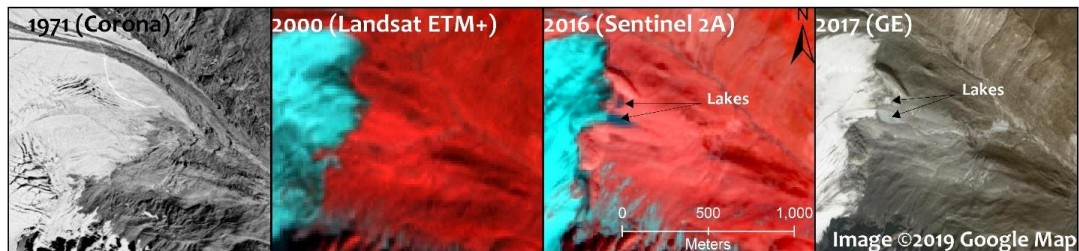

(b) Field photograph (2016) of pro-glacier lakes at the terminus zone of G56 glacier.

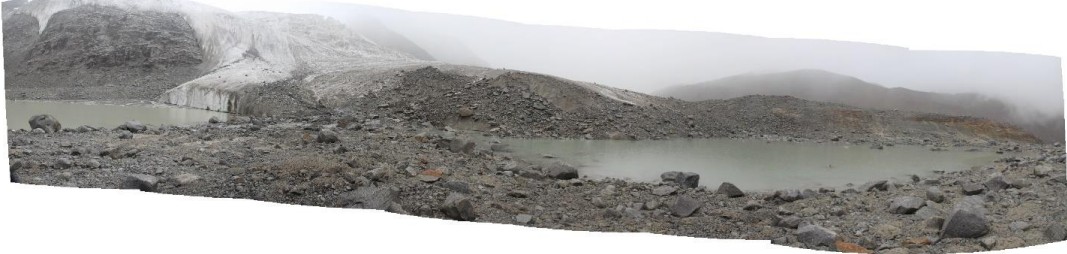

**Figure 3**. Terminus characteristics of G56 glaciers in the JCW. (a) two proglacial lakes were evolved in the terminus zone of G56 glacier during 2000–16. (b) Field photographs of proglacial lakes of G56 glacier (2016).

### 4.2 Glacier length change measurements and related uncertainty

Glacier length has been calculated based on criteria suggested by Lopez et al. (2010) for 127 glaciers. Glacier length was represented by a line (mainly center flow line) which corresponds to the longest flow distance of a glacier (Fig. 4a). Source of the central position of the terminus and surface flow trajectories of glaciers were considered as are identifiable on satellite images. All criterion was applied to each glacier except surface trajectories, which depend on glacier shape and surface morphology (Lopez et al., 2010). Multiple accumulation areas, complex geometry and highly crevassed terminus with debris cover on G34, G54, G56, and G68 glaciers made it hard to delineate the central line. High-resolution GE images and field measurements/mapping have been employed to overcome these uncertainties. For the calculation of length changes, stripes with 50 m distance were drawn parallel to the main flow direction of the glacier (Koblet et al., 2010). Length change was calculated along the central flow line to compare the result with the derived average length change (Fig. 4b1). Besides, Length change was calculated as the average length from the intersection of the stripes with multitemporal glacier outlines (Fig. 4b2). Terminus retreat of nine select glaciers was monitored in detail.

Sensor resolution and co-registration errors limit the accuracy of the measurement of the front position of glaciers. Terminus uncertainty (TU) is calculated using the following formula reported by Bhambri et al. (2012):

$$TU = \sqrt{[(PR^a)^2 + (PR^b)^2]} + E_{reg} \qquad (1)$$





Where $PR^a$ is the pixel resolution of imagery 1, $PR^b$ is the pixel resolution of imagery 2, and $E_{reg}$ is the registration

error. The uncertainty can be estimated in the case of Corona imagery (1971) as follows:

$$TU = \sqrt{[(2)^2 + (10)^2]} + 5.6 = 15.8\ m \qquad (2)$$

The uncertainty was calculated at ~28.5 m for Landsat pan-sharpened ETM+ image (2000).

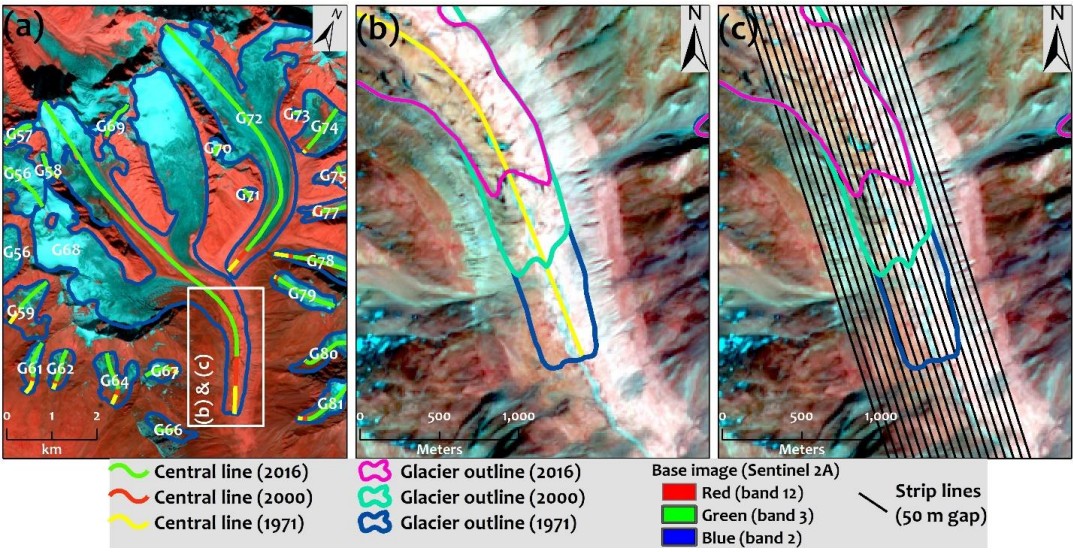

**Figure 4.** (a) Example of length measurement of the G68 (G077050E32943N) and its surroundings glaciers along the central flow line. (b) Length change measurements based on central flow line for the G68 glacier. (c) Length change measurements based on 50 m spacing horizontal strip lines. The background image is Sentinel 2A false-color composite (bands: 12–3–2).


### 4.3 DEMs processing and related uncertainties

#### 4.3.1 SRTM DEM: radar penetration correction

Previous studies have shown that SRTM C-band penetrates more relative to the X-band (Gardelle et al., 2013; Vijay and Braun, 2016; Mukherjee et al., 2018; Kääb et al., 2012; Gardelle et al., 2012). For comparison between SRTM C-/X-

band DEMs, we obtained 10° x 10° mosaics of X-band DEM (tile id: E070N30). Unfortunately, no X-band DEM is available for the JCW (Fig. S1). A recent comprehensive study by Vijay and Braun (2016) showed elevation-dependent penetration measurements of SRTM C-band in the adjacent region of the JCW (Fig. S1). We adjusted SRTM C-band DEM using their derived radar penetration measurements (See section 5.1 of Vijay and Braun (2016) for detail). Also, Mukherjee et al. (2018) reported similar results for the same region.

#### 4.3.2 Co-registration of DEMs, elevation changes and data gaps

We used a robust analytical 3-D co-registration method proposed by Nuth and Kääb (2011) to remove the horizontal and vertical shifts between the DEMs. This procedure adjusts a reference DEM (TanDEM in our case) iteratively



concerning a base DEM (SRTM) until a minimum standard deviation of the elevation difference is reached in the ice-free terrain (Fig. 5). Before co-registration of the DEMs, 30 m SRTM C-band DEM was resampled to 90 m followed

by a vertical datum transformation of TanDEM to the similar of SRTM DEM (i.e., EGM96 geoid; Gardelle et al., 2012).

Once the DEMs were co-registered, we calculated glacier elevation changes by subtracting the newer TanDEM to the older SRTM DEM. We created an off–glacier mask based on the slope threshold of <25°. For the stable terrain, outliers were defined by 1.5 fold of interquartile range (after Pieczonka and Bolch (2015)). Finally, outliers for the glacierized

area were defined as (a) surface elevation changes more than ± 30 m or (b) elevation difference greater than mean plus three standard deviations of the stable terrain elevation bias (Gardelle et al., 2013; Robson et al., 2018). Later, data gaps were filled using ordinary kriging (Pieczonka and Bolch, 2015).

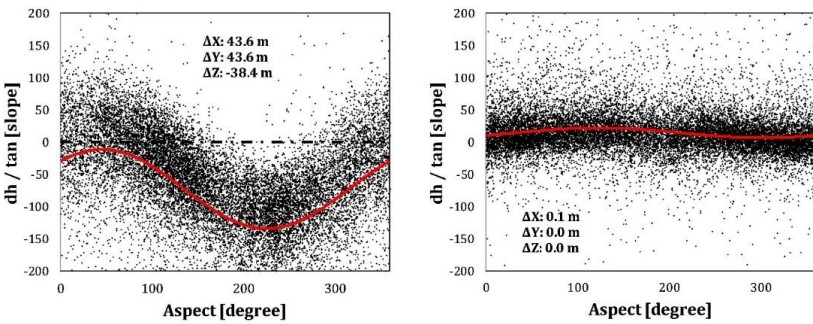

**Figure 5.** Plots of slope normalized terrain elevation differences between the 2000 SRTM and the TanDEM–X (2014) DEM over stable terrain (off glacier) before co-registration (left) and after five iterations of co-registration (right).

### 4.3.3    Geodetic mass budget estimation

First, the volume change (V) was calculated by integrating the mean elevation change (*dh*) values with the

corresponding glacier area for individually analyzed glaciers. Later, volume change was converted to the mass change by multiplying by a density conversion factor of 850 ± 60 kg m$^{-3}$ ($\rho_{ice}$) (Gardelle et al., 2013; Huss, 2013). The geodetic mass budget was calculated using glacier outlines of 2016. The final geodetic mass balance ($\dot{B}$) was calculated using the following formula:

$$\dot{B} = \frac{(dh \times A_{surf}) \times \rho_{ice}}{A_{surf} \times \rho_w} \tag{3}$$

Where, $A_{surf}$ is the total glacier area for a given glacier, and $\rho_w$ is the water density (999.972 km m$^{-3}$).

### 4.3.4    Error assessment



In the present study, surface elevation change uncertainty mainly arises due to the uncertainty of DEM difference images ($U_{dh}$), the uncertainty of radar penetration ($U_{RP}$) and the uncertainty of the glacier area change ($U_{\Delta surf}$). The

uncertainty measurement of DEM difference was calculated using the Eq. (4) to (6) according to Gardelle et al. (2013).

$$U_{dh} = \frac{1}{n}\sum_{i=1}^{n} U_{dh}(i) \qquad (4)$$

$$\text{Where, } U_{dh}(i) = \frac{\delta_{dh}(i)}{\sqrt{N_{eff}}} \qquad (5)$$

$$\text{And, } N_{eff} = \frac{N_{tot(i)} \times dh_{PR}}{2D} \qquad (6)$$

Where, $U_{dh}(i)$ is the uncertainty corresponding to the i'th altitude band, $\delta_{dh}(i)$ is the standard deviation of the mean

elevation change of the stable terrain of the i'th altitude band, $N_{tot(i)}$ is the total number of pixels in the i'th altitude band, $dh_{PR}$ is the pixel resolution of the elevation difference image, and $D$ is the spatial autocorrelation distance (in meters) of elevation change in stable terrain. Spatial autocorrelation was determined using Moran's I autocorrelation index off–glacier mask.

The uncertainty of the radar penetration correction was used as 1.41 m as reported by Mukherjee et al. (2018). The

overall uncertainty of thickness changes ($U_{EC}$) was calculated using the following formula of standard error propagation mentioned by Mukherjee et al. (2018):

$$U_{EC} = \sqrt{(U_{dh})^2 + (U_{RP})^2 + \left(U_{\Delta surf}\right)^2} \qquad (7)$$

The uncertainty of density of ice ($U_{\rho_{ice}}$) was considered as $\pm 60$ kg m$^{-3}$ for the mass budget measurement. Finally, the mass budget uncertainty ($U_M$) was calculated using the following Eq.

$$U_M = \sqrt{\left(\frac{\Delta h . U_{\rho_{ice}}}{t . \rho_w}\right)^2 + \left(\frac{U_{EC} . \rho_{ice}}{t . \rho_w}\right)^2} \qquad (8)$$

Where, $\Delta_h$ is the surface elevation difference, and t is the time period in years.

### 4.4    Glacier surface velocity and related uncertainty

#### 4.4.1    Landsat images pair cross-correlation and pre-processing

Glacier surface displacements were determined using normalized cross-correlation algorithm from multitemporal

Landsat images (TM, ETM+, OLI) (Leprince et al., 2007, 2008). The method is compiled in a software package known as the Co-registration of Optically Sensed Images and Correlation (COSI–Corr). This is a free plugin for the ENVI software (Ayoub et al., 2009). COSI–Corr is widely used to measure the glacier surface velocities for push-broom sensors like SPOT and ASTER. Consecutive cloud-free ASTER scenes of the ablation period are sporadic for the JCW. Landsat data have some advantages over ASTER. Landsat images have a large footprint as compared to ASTER

and are available since the 1970s. Landsat images are acquired vertically (at nadir without along and across track pointing) so that they contain minimal topographic distortions. However, Scherler et al. (2008) reported subpixel noise on Landsat data created by unknown attitude variations of the satellites. Nevertheless, the horizontal accuracy of <~6





m has been reported for Landsat TM, and ETM+ sensor (Tucker et al., 2004) which can be considered within the acceptable limit as most glacial features exceed this noise level (Bhattacharya et al., 2016).

The algorithm available in COSI–Corr works on single-band greyscale images. In this study, we evaluated the panchromatic band (PAN; band 8; 15 m; 0.50–0.66 μm for OLI and 0.52–0.9 μm for ETM+) for Landsat OLI and ETM+ and green band (band 2; 30 m; 0.52–0.6 μm) for Landsat TM. The green band of Landsat TM was chosen as its wavelength is close to the PAN band of Landsat OLI and ETM+. The images were preprocessed based on the method used by Berthier et al. (2003). At first, the principal component of the bands was computed. This step yields

low noise and enhances the topography. After that, a convolution spatial filter using a kernel size of 3 x 3 was applied to generate the high–pass filtered image sets. These image sets were used to measure displacement. Only the glaciers >1 km² in size in 2016 (33 glaciers) were used for surface velocity measurements as the small morphological features (i.e., mainly crevasses, flow patterns) are identifiable on these glaciers based on satellite images.

### 4.4.2 COSI–Corr: extraction of glacier surface displacements

COSI–Corr provides two correlation algorithms: (1) frequency correlation and (2) statistical correlation (Leprince et al., 2007). The frequency correlation is Fourier based and is more accurate than the statistical one (Ayoub et al., 2009). This correlation is more sensitive to noise and is therefore recommended for optical images of good quality. The statistical correlation maximizes the absolute value of the correlation coefficient and is coarser but more robust than the frequential one. Its use is recommended for correlating noisy optical images that provided terrible results with the

frequency correlation (Leprince et al., 2007). We used the frequency correlation algorithm for glacier displacement measurement in the JCW. The input images should be co-registered as accurately as possible as the correlation algorithm is sensitive to image misregistration errors. Besides, image noise due to stripping, sensor noise, and illumination differences caused by cloud and shadow will have a deteriorating effect on displacement results (Scherler et al., 2008; Lucieer et al., 2014).

The frequency correlation algorithm in COSI–Corr requires a number of initial parameter settings (Ayoub et al., 2009): (1) window size – the size in pixels that will be correlated; (2) step – determines the step in pixels between two sliding windows; (3) robustness iteration – the number of times per measurement the frequency mask should be recomputed; and (4) mask threshold – the masking of the frequency according to the amplitude of the log-cross spectrum. A typical correlation analysis was performed using a 64 (initial) and 32 (final) pixel window size and a 2-

pixel step. The robustness iteration and mask threshold were assigned to 2 and 0.9, respectively. Appropriate settings of these values need a priori field-based knowledge of the size of the displacements, which is not available for the JCW. The correlation algorithm results in the three output images. The first two images provide the 2D displacements in terms of the east-west displacement (EWD; east positive) and the north-south displacement (NSD; south positive) both expressed in meters (Lucieer et al., 2014). The ground displacement (D) was measured by combining these two

images by the square root of the sum of the squared displacements (Eq. 9). The surface velocity (V) was calculated by dividing the displacement with a time interval (T) between two images (Eq. 10). The third image shows the signal–to–noise ratio (SNR) helps to assess the quality of the displacement image. An example of glacier surface velocity measurement is presented in Fig. 6.

$$D = \sqrt{(EWD)^2 + (NSD)^2} \qquad\qquad (9)$$





$V = D/t$                          (10)

### 4.4.3     Post–processing and error assessment

The post-processing of derived displacement maps was performed to remove the spurious and false values using two sequential filters (Berthier et al., 2005; Scherler et al., 2008). At first, a filter was applied to exclude poorly correlated pixels (SNR < 0.950). Secondly, a magnitude filter was applied to remove the pixels showing isolated very high

displacement values. In our case, we manually examined the regions of maximum high velocity on the glacier surface. We assumed the maximum realistic surface speed was < 60 m a$^{-1}$ in the JCW. Values above this limit were removed from the measurements. Besides, field-based measurements using stakes in the Menthosa glacier (76°44'29.44"E; 32°54'31.65"N) in the adjacent Miyar valley also supports this assumption. Field measurements on Menthosa glacier were carried out under the Department of Science and Technology, Government of India, sponsored project entitled

as "Himalayan Cryosphere: Science and Society" between 2013 and 2018 (personal communication with project investigator Prof. Milap Chand Sharma). As a result, most exceptional values due to poor correlation, shadow, and cloud cover were discarded from the displacement measurement. Mean velocity measurements were calculated based on the glacier outline of 2016 and the central line drawn along the main flow direction.

The uncertainty in surface displacements using cross-correlation techniques generates from basically three sources:

(1) the image co-registration process, (2) the quality of the scenes, and (3) the performance of the cross-correlation algorithm. All these sources combinedly affect the accuracy of the velocity measurements. We estimated the velocity uncertainty by measuring the total displacement on stable terrain (<25° slope) based on previously developed methods (Scherler et al., 2008; Berthier et al., 2005). This resulted in an uncertainty of < 5 m a$^{-1}$ for all observation periods (Table S3).

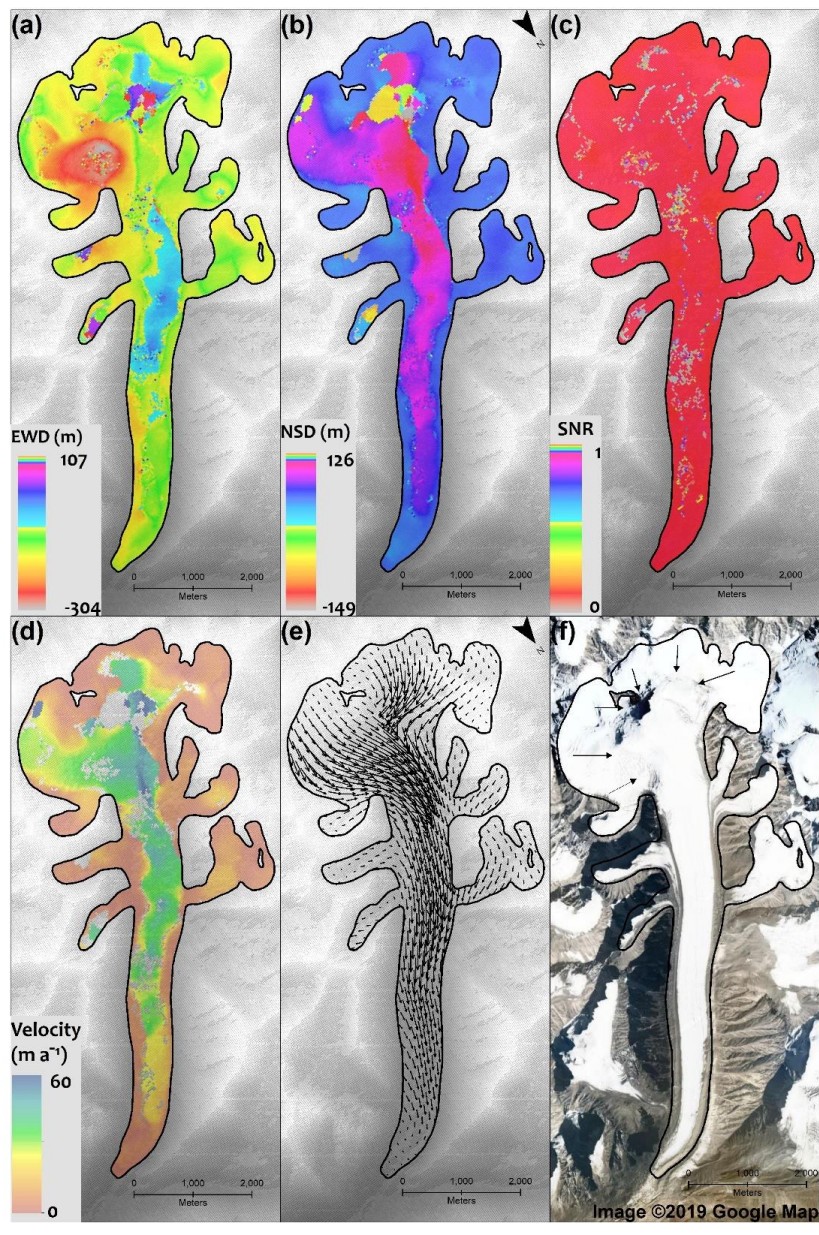

**Figure 6.** An example of surface velocity measurements of G54 glacier (G076984E32862N) in the JCW based on normalized cross-correlation of Landsat 8 satellite images using COSI–Corr software during 2016–17. (a) East-west displacement (EWD) map. (b) North-south displacement (NSD) map. (c) Signal–to–noise ratio (SNR) map. (c) Final surface velocity map data gaps after threshold adjustment. (e) Velocity vectors. (f) High-resolution Google Earth image. The black arrow in (f) represents the zone of high crevasses and maximum surface slope.

## 5    Results

### 5.1    Terminus retreat

The present study reveals a heterogeneous terminus retreat in all glaciers in the JCW during the study period (1971–
2016) (Table S2). Mean terminus retreat ranges from ~4 ± 15.8 m (0.1 ± 0.4 m a$^{-1}$; G13) to 1274 ± 15.8 m (28.3 ± 0.4
m a$^{-1}$; G68) with an average retreat rate of 4.7 ± 0.4 m a$^{-1}$ during the analysis period. Out of the total 127 glaciers,
only one glacier had a terminus change of > 1000 m (G68: ~1274 m), whereas 80 glaciers receded < 200 m between
1971 and 2016 (Table S2). Terminus retreat of some of the select glaciers is given in Fig. 7 and Table 2. Glaciers
between 5 and 10 km² in size show lowest retreat rate of 0.2% a$^{-1}$. In contrast, smaller glaciers (< 0.5 km² in size)
retreated at a faster rate (~0.4% a$^{-1}$). Furthermore, clean-ice glaciers retreated at a faster rate (~0.3% a$^{-1}$) as compared
to debris-covered ones (~0.2% a$^{-1}$).

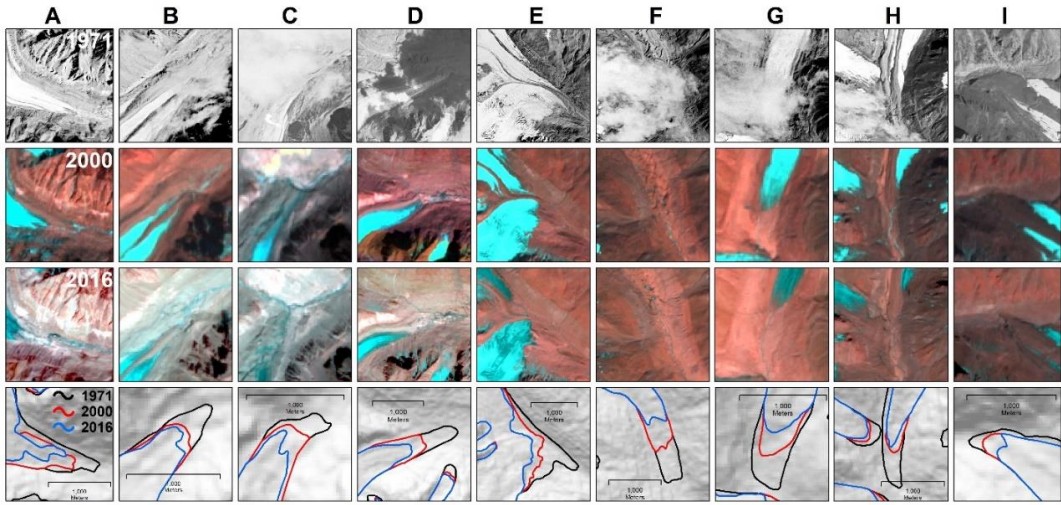

**Figure 7.** Terminus change of nine select glaciers in the JCW between 1971 and 2016. (A) Kelas Buk glacier (G12;
G077065E32681N); (B) Chhallopathgang glacier (G20; G077079E32731N); (C) Dali Chhu glacier (G34;
G077027E32775N); (D) G54 glacier (G076984E32862N); (E) G56 glacier (G076984E32920N); (F) Bagrari Chhu
glacier (G68; G077050E32943N); (G) G72 glacier (G077041E32979N); (H) G87 glacier (G077116E32952N); and
(I) G114 glacier (G077129E32873N).

### 5.2    Elevation change and mass budget

We quantified elevation change for 127 glaciers in the JCW with a total area of ~181.4 ± 3.6 km² during 2000–14.
The spatial distribution of glacier elevation change is presented in Fig. 8. We observed elevation gain for nine glaciers
(ranges from 0.1 to 0.7 m a$^{-1}$) while the rest of the glaciers show surface lowering (ranges from –0.1 to –1.5 m a$^{-1}$)
(Table S2). The investigated glaciers in the JCW thinned on average of –0.7 ± 0.4 m a$^{-1}$, leading to a mass loss of –
0.6 ± 0.4 m w.e. a$^{-1}$ (Table 2). The mass budget of individuals glaciers varies between –1.31 and –0.6 m w.e. a$^{-1}$.
Figure 9 and Table 2 show the surface lowering and geodetic mass balance of the entire glacierized area of the JCW
and nine select glaciers during 2000–14. The largest glacier in the JCW (G34; G077027E32775N; ~21.7 km²) thinned


at a rate of –0.8 ± 0.4 m a$^{-1}$ with mass loss of –0.7 ± 0.3 m w.e. a$^{-1}$. Mass budget estimates of select nine glaciers were slightly higher than the mean of the JCW (Table 2).


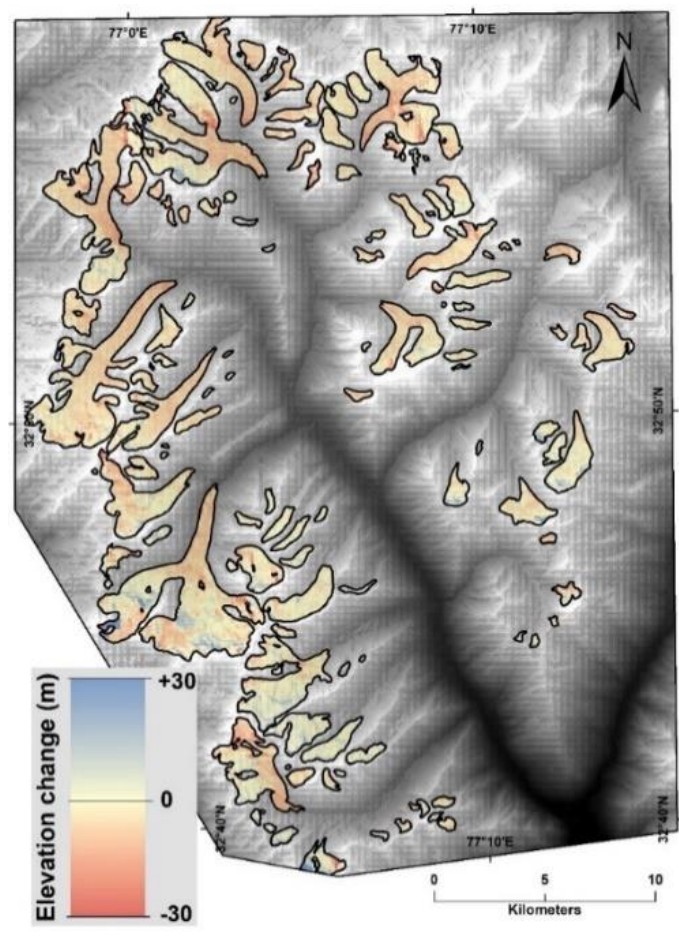

**Figure 8.** Map of elevation change of glaciers in the JCW between 2000 (SRTM C$_{-band}$ DEM) and 2014 (global TANDEM X$_{-band}$ DEM). The hill shade map with 30% opacity derived from SRTM 30 m global DEM is used as background image.



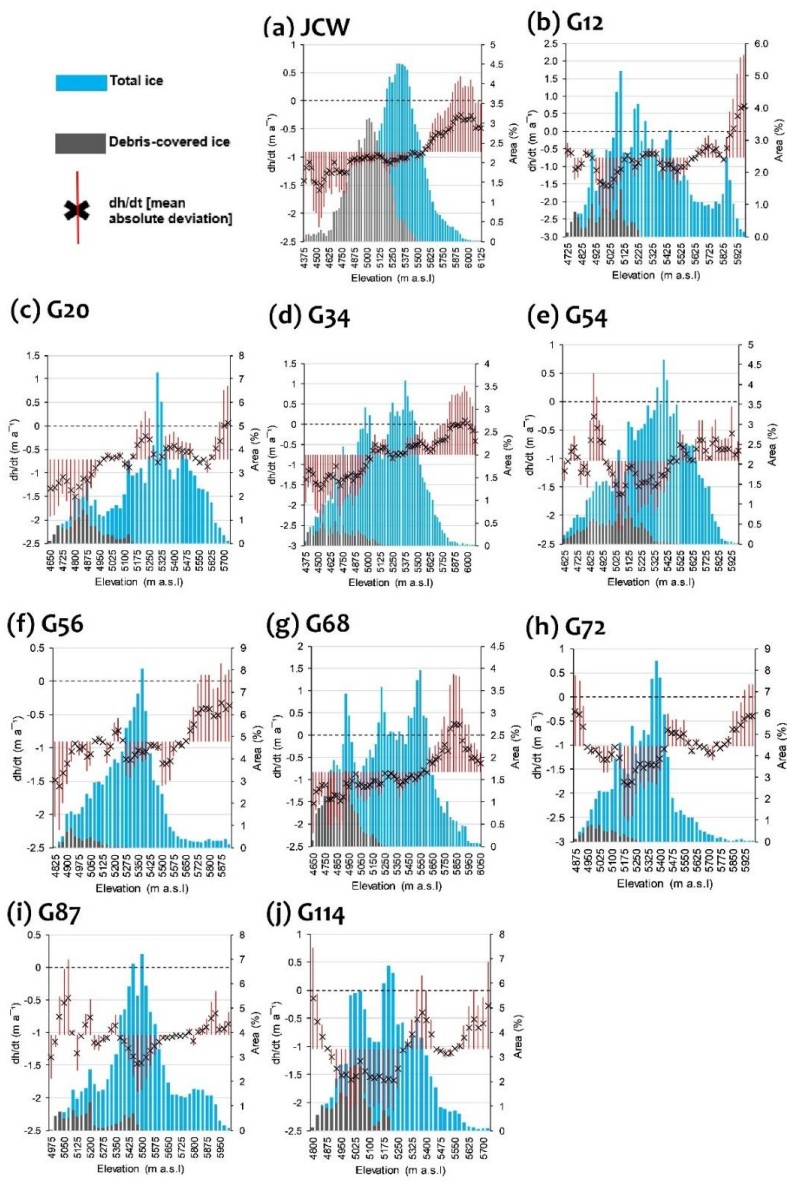


**Figure 9.** The hypsometric elevation change plots of the total and nine select glaciers in the JCW. The elevation bin is derived at an interval of 25 m. (a) The Jankar Chhu Watershed. (b) Kelas Buk glacier (G12; G077065E32681N); (c) Chhallopathgang glacier (G20; G077079E32731N); (d) Dali Chhu glacier (G34; G077027E32775N); (e) G54 glacier (G076984E32862N); (f) G56 glacier (G076984E32920N); (g) Bagrari Chhu glacier (G68; G077050E32943N); (h) G72 glacier (G077041E32979N); (i) G87 glacier (G077116E32952N); and (j) G114 glacier (G077129E32873N).






### 5.3 Surface velocity

The velocity measurements using COSI–Corr show that glacier in the JCW is active throughout the observation period (1993–2016). The surface velocity map for the JCW is presented in Fig. 10. A details velocity measurement for analyzed 33 glaciers are given in Table S4. Most of the data gaps mainly occur in the upper accumulation zone due to the lack of visible surface features. The lower part of the glaciers shows a very consistent result with tiny data gaps. Mean annual surface velocities during 2000/02 were higher (average velocity ~9.42 m a$^{-1}$) as compared to the other periods. The annual surface velocity during 2009/11 was lowest (~3.25 m a$^{-1}$) followed by 1993/94 (~5.44 m a$^{-1}$). In 2016/17 period, the mean velocity was measured at ~8.46 m a$^{-1}$. Velocities derived along the central flow line show a remarkable similarity with mean velocity derived using glacier polygons (Table S4). The annual velocity on clean glaciers was observed to be ~35% higher than debris-covered ones between 2016/17 and 2000/02 period.

Velocity measurements of select nine glaciers show that G114 glacier speed was highest (~18.95 m a$^{-1}$) while G12 glacier speed was lowest (~5.46 m a$^{-1}$) during the 2016-17 period (Table 2). The annual surface velocity of G34 glacier was higher (~14.5 m a$^{-1}$) in 2000/02 and 2016/17 as compared to the 1993/94 (~9.88 m a$^{-1}$) and 2009/11 period (~4.11 m a$^{-1}$). Our velocity measurements show that surface velocity was higher during 2000/02 and 2016/17 as compared to the other two observation periods for all the analyzed glaciers in the JCW.



**Figure 10.** Glacier surface velocity maps of four different periods for the Jankar Chhu Watershed.




**Table 2.** Glacier surface area change, terminus retreat, and mass balance results for the entire JCW and nine select glaciers. L = length, ΔL = length change, ΔElev = elevation change, and Derb$_{cover}$ = debris cover. V$_{Mean}$ = mean velocity based on glacier outline; V$_{ACL}$ = Velocity along the central line. Velocity uncertainty was measured at <5 m a$^{-1}$.

| | | Terminus retreat | | | | | |
| | | Along the central line | | | | Based on horizontal strip lines | |
| Glacier ID | GLIMS ID | L$_{1971}$(km) | L$_{2016}$(km) | ΔL$_{1971\text{-}2016}$ (m) | ΔL$_{1971\text{-}2016}$ (m a$^{-1}$) | ΔL$_{1971\text{-}2016}$ (m) | ΔL$_{1971\text{-}2016}$ (m a$^{-1}$) |
|---|---|---|---|---|---|---|---|
| G12 | G077065E32681N | 5.4 | 4.9 | -500.6 | -10.1 | -540.6 | -12.2 |
| G20 | G077079E32731N | 5.4 | 5 | -406.5 | -9.4 | -406.3 | -9 |
| G34 | G077027E32775N | 8.3 | 7.7 | -607.5 | -12.7 | -526.3 | -11.7 |
| G54 | G076984E32862N | 10.2 | 9.2 | -996.2 | -21.2 | -979 | -21.8 |
| G56 | G076984E32920N | 8.3 | 6.5 | -1786.5 | -40.4 | -738.5 | -16.4 |
| G68 | G077050E32943N | 8.5 | 7.3 | -1308.4 | -28.3 | -1202.5 | -26.7 |
| G72 | G077041E32979N | 6.7 | 6.1 | -587.6 | -13.4 | -540.7 | -12 |
| G87 | G077116E32952N | 6 | 5.4 | -615.8 | -14.5 | -603.2 | -13.4 |
| G114 | G077129E32873N | 4.8 | 4.5 | -280.5 | -7.6 | -231.7 | -5.1 |
| JCW | | 1.8 | 1.6 | -285.7 | -5.3 | -206.4 | -4.6 |

| | | Elevation change and mass budget | | | | | |
| | | ΔElev (m) | ΔElev (m a$^{-1}$) | Volume change (10$^{-3}$ km$^3$ a$^{-1}$) | Mass balance (m w.e. a$^{-1}$) | Derb$_{cover}$ (%) | Slope (°) |
|---|---|---|---|---|---|---|---|
| G12 | G077065E32681N | -3.6 ± 4.9 | -0.3 ± 0.4 | -6.1 ± 2.4 | -0.8 ± 0.3 | 12 | 18 |
| G20 | G077079E32731N | -9.5 ± 4.9 | -0.7 ± 0.4 | -5.6 ± 2.9 | -0.6 ± 0.3 | 15 | 18 |
| G34 | G077027E32775N | -11.7 ± 4.9 | -0.8 ± 0.4 | -18.2 ± 7.7 | -0.7 ± 0.3 | 8 | 18 |
| G54 | G076984E32862N | -16.0 ± 4.9 | -1.1 ± 0.4 | -17.0 ± 5.3 | -1.0 ± 0.3 | 3 | 17 |
| G56 | G076984E32920N | -14.2 ± 4.9 | -1.0 ± 0.4 | -13.6 ± 4.8 | -0.9 ± 0.3 | 4 | 17 |
| G68 | G077050E32943N | -13.5 ± 4.9 | -1.0 ± 0.4 | -12.9 ± 4.7 | -0.8 ± 0.3 | 19 | 17 |
| G72 | G077041E32979N | -17.9 ± 4.9 | -1.3 ± 0.4 | -8.5 ± 2.4 | -1.1 ± 0.3 | 12 | 15 |
| G87 | G077116E32952N | -16.0 ± 4.9 | -1.1 ± 0.4 | -9.1 ± 2.8 | -1.0 ± 0.3 | 12 | 19 |
| G114 | G077129E32873N | -17.6 ± 4.9 | -1.3 ± 0.4 | -6.2 ± 1.7 | -1.1 ± 0.3 | 20 | 14 |
| JCW | | -10.4 ± 4.9 | -0.7 ± 0.4 | -129.8 ± 47.3 | -0.6 ± 0.3 | 12 | 23 |

| | | Surface velocity (m a$^{-1}$) | | | | | | | |
| | | 1993–1994 | | 2000–2002 | | 2009–2011 | | 2016–2017 | |
| | | V$_{Mean}$ | V$_{ACL}$ | V$_{Mean}$ | V$_{ACL}$ | V$_{Mean}$ | V$_{ACL}$ | V$_{Mean}$ | V$_{ACL}$ |
|---|---|---|---|---|---|---|---|---|---|
| G12 | G077065E32681N | 6.1 | 10.7 | 10 | 15.8 | 3.9 | 7 | 6 | 5.3 |
| G20 | G077079E32731N | 6.7 | 6.9 | 12.1 | 14.2 | 2.8 | 3.3 | 11.4 | 15.3 |
| G34 | G077027E32775N | 9.9 | 17.1 | 14.4 | 23.2 | 4.1 | 7.9 | 14.7 | 20.4 |
| G54 | G076984E32862N | 9.2 | 14.8 | 10.6 | 20.4 | 4.9 | 6.5 | 13.9 | 25.4 |
| G56 | G076984E32920N | 5.2 | 7.2 | 9.7 | 11.1 | 3.9 | 5.8 | 7.4 | 11.1 |
| G68 | G077050E32943N | 6.3 | 8.5 | 11.4 | 14.9 | 3.8 | 5.8 | 7.4 | 4.7 |
| G72 | G077041E32979N | 3.9 | 5.5 | 7.6 | 12.2 | 2.1 | 2.2 | 6.3 | 7.8 |
| G87 | G077116E32952N | 4.3 | 5.2 | 8.2 | 8.1 | 2.4 | 2.4 | 6.9 | 6.5 |
| G114 | G077129E32873N | 11.6 | 9.1 | 12.7 | 15.4 | 6 | 7.3 | 18.9 | 20 |
| JCW | | 5.4 | 6.4 | 9.4 | 12.3 | 3.2 | 3.9 | 8.5 | 10.3 |




## 6 Multiparameter glacier change dynamics in the JCW: A comparative analysis

Glaciers in general, in the JCW receded by ~211 ± 15.8 m (~4.7 ± 0.4 m a⁻¹) between 1971 and 2016. In adjacent eastern Chandra basin, Pandey and Venkataraman (2013) investigated length change of 15 select glaciers. They suggest that the mean glacier terminus retreated by 465.5 ± 169.1 m during 1980–2010 with an average rate of 15.5 ± 5.6 m a⁻¹. Schmidt and Nüsser (2017) have reported that glaciers in Kang Yatze massif receded by about 125 m (~3 m a⁻¹) between 1969 and 2010, is similar to our results. Also, in the Garhwal and Kumaon Himalaya, the frontal retreat rate is found to be more than 20 m a⁻¹ (Bhambri et al., 2012).

Several studies have computed the geodetic mass balance of glaciers in the adjacent regions of the JCW, Lahaul Himalaya in different periods (Kääb et al., 2012; Berthier et al., 2007; Gardelle et al., 2013; Vijay and Braun, 2016; Mukherjee et al., 2018; Zhou et al., 2018). Kääb et al. (2012) reported the geodetic mass balance of –0.3 ± 0.1 m w.e. a⁻¹ (750 kg m⁻³ ice density assumption) for glaciers in the Lahaul–Spiti and adjacent regions during 2003–09. The difference in present mass balance estimate (–0.6 ± 0.3 m w.e. a⁻¹) can be primarily attributed to different ice density assumptions and periods of observation. Between 1999 and 2004, Berthier et al. (2007) measured geodetic mass balance of –0.7 to –0.85 m w.e.a⁻¹ in the adjacent Bara Shigri group of glaciers. We obtained similar results for the JCW. During 1999–2011, Gardelle et al. (2013) estimated geodetic mass balance of –0.45 ± 0.14 m w.e.a⁻¹ for the glaciers in the Lahaul–Spiti region based on SRTM and SPOT5 datasets with similar ice density assumption (850 kg m⁻³). Recent comprehensive study based on SRTM and high-resolution TanDEM data, Vijay and Braun (2016) reported a geodetic mass balance of –0.53 ± 0.37 m w.e.a⁻¹ during 2000–12 with similar ice density assumption to our study. Our results suggest a similar mass loss rate in line with the observation of Vijay and Braun (2016). Besides, Mukherjee et al. (2018) reported a slightly lower mass loss in a similar region of Lahaul–Spiti. However, the main uncertainty related to our results mainly comes from the use of low-resolution TanDEM (90 m).

We present the first surface velocity measurements for the JCW of the Greater Himalayan range based on remote sensing methods. We observed a low mean surface velocity (<15 m a⁻¹) probably because of the use of a comparatively lower maximum realistic surface displacement threshold value (± 60 m). This threshold value also confirmed by the filed observation in Menthosa glacier (a benchmark glacier) in the adjacent basin of Miyar valley using stakes measurements in ablation and accumulation zone since 2014. We observed that highly debris-covered ablation zone of Menthosa glacier displaced at a rate of <6 m a⁻¹ while accumulation zone displaced at a rate of <20 m a⁻¹. We assume that the glaciers in the JCW also behave similarly by taking into consideration of similar glacier morphology and topographic settings in both valleys. Spatial variability of glacier surface speed among individual glaciers appears to relate to the altitudinal range broadly, amount of debris cover, shape and slope and hypsometry of glaciers (Scherler et al., 2008, 2011b; Robson et al., 2018). There is also considerable temporal and spatial variation in velocity within the individual glaciers. Also, the difference in spatial resolution between satellite images and seasonal snow cover can lead to erroneous velocity measurements.

## 7 Data availability

The dataset for multiparameter and multi-temporal glacier change dynamics in the JCW, Lahaul Himalaya is freely available at http://doi.org/10.5281/zenodo.3383233 (Das and Sharma, 2019b). Field photographs can be obtained from the corresponding author with a reasonable request.



## 8    Conclusion

The study focused on the systematic assessment of 127 glaciers in the JCW, Lahaul Himalaya between 1971 and 2016 based on remote sensing. The newly derived multiparameter glacier change dataset presented here should be of the general interest of Earth System Science, more particularly for those studying glacier fluctuations/modeling in the Himalayan region. We foresee that it could become a valuable asset to understand the regional climate glacier interactions. Further, the Global Land Ice Measurements from Space (GLIMS) and the International Centre for

Integrated Mountain Development (ICIMOD) database could be enriched by incorporating present datasets to their server. The following significant inferences are drawn from the analysis:

- Average terminus retreat ($4.7 \pm 0.4$ m a$^{-1}$) is observed to be much lower than previously reported.
- The investigated glaciers in the JCW thinned on average of $-0.7 \pm 0.4$ m a$^{-1}$, leading to a mass loss of $-0.6 \pm 0.4$ m w.e. a$^{-1}$, which is similar to the findings of other studies in the adjacent basins.

- For the first time, the surface velocity of glaciers was determined in the JCW. Mean annual surface velocities during 2000–02 were higher (average velocity ~9.42 m a$^{-1}$) as compared to the other periods. During the 2016–17 period, the mean velocity was measured at ~8.46 m a$^{-1}$.

### Supplementary

### Author contributions

S. Das collected, processed, and analyzed the data and wrote the original draft of the manuscript. MCS conceptualized and supervised the study and jointly wrote the manuscript with S. Das.

### Acknowledgment

S. Das is thankful to the University Grant Commission, New Delhi (3090/ (NET–DEC.2014) for financial support during field observations. S. Das thank Satish and Vikesh of Miyar Valley, Lahaul, for their help during field visits.

S. Das also express thanks to Etienne Berthier for freely providing the DEM co-registration and velocity mapping tool (http://etienne.berthier.free.fr/tutorial.htm). We thank USGS for providing Sentinel 2A, Landsat, Corona, and SRTM C–band data at no cost. We also acknowledge the Earth Observation Center (EOC) of the German Aerospace Center (DLR) for freely providing TanDEM-X and SRTM X–band data.

### Funding

The author(s) received no financial support for authorship and publication of this article.

### Conflicting of Interests

The author(s) declared no potential conflicts of interest concerning the research, authorship, and publication of this article.

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
