# Peer review of "Glacier terminus retreat, mass budget, and surface velocity measurements for the Jankar Chhu Watershed, Lahaul Himalaya, India"

_Earth System Science Data, 2019_

## Referee Comment (RC1) · Anonymous Referee #1 · 7 Feb 2020

The authors describe the combination and production of several remote sensing datasets, together with some fieldwork observations from a valley in the Himalaya. Apart from some minor technical implementations, this work seems like common practice. At first hand this sounds like a positive point, but it can also be seen as a weakness. It raises the point what exactly the authors want to convey with this dataset. Their study starts with a review on literature on glacier change in the Himalaya, with a big table in the supplement. There is a section with some temperature measurements and debris cover thickness. And their work as presented in 2019 about glacier outlines is explained again in this work.

In my opinion, the new contribution introduced in this study comes from the remote sensing datasets. The used datasets are generally available, such as SRTM and TanDEM-X, and these have been used for the whole mountain range of the HKH. Similarly, datasets of velocity are available in (annual) mosaic form (such as ITSLIVE) or individual pairs (like GOLIVE). These are similar datasets, but rely on multi-temporal velocity estimates (in the case of ITSLIVE), while the authors rely on single data pairs. Consequently, the velocity results do not seem to be physically reasonable (as large blobs of velocity exist in all shown time periods). The estimated precision might be in line with the real uncertainty, but already by eye gross errors are present, so there is an issue with reliability. I can not see why such a dataset would be of interest. Furthermore, the authors produce a mean velocity over a glacier, while it has gaps (thus a changing sampling set) and velocity is highly dependent on slope.

Concerning the elevation changes, the results presented in Figure 8, seem to have an aspect dependency. Many north facing slopes have a positive elevation change, this might be a real signal, but for example glacier G18 experiences area loss, while a general elevation gain is measured. This contradicts each other, especially in the lower snout. Such errors will be part of the noise for regional estimates, but the authors use it here on an individual glacier basis. Again, if sufficient temporal data is present, these errors might diminish, but this is not the case here. Then data from HMA from World-View is such a dataset and very helpful (https://nsidc.org/data/highmountainasia). Given in light of these available and open datasets it is very difficult to distill what the aim is of this dataset. Especially, because validation data is not present, which is difficult for remote sensing of high mountain glaciers. But, given the detail the authors aim for here, and the envisioned audience, this is a strong shortcoming.

The main points of this study are given in the conclusion, being - area change, which was highlighted by the authors in already another paper - elevation change, which was done on a large scale by others as well - velocity change, which was also already done

Furthermore, there are large sections about SRTM C and X band penetrations, while

only C-band data is used. In addition COSI-CORR is explained, but not correctly and too lengthy. The reader might also wonder why the temperature measurements are given, but not placed into context.

Given below are some minor remarks: p01.l01.: title not correct, maybe change to "Remote sensing of ...". "Mass budget" is not given change to "elevation change", "measurements"... "estimates" might be more suited.

p01.l13.: "and the limited" > "and a limited"

p01.l20.: "but the dynamic active" what is meant here?

p01.l21.: which "findings" do you mean?

p01.l29.: "enourmous" please use appropriate wording, be less subjective "large" might be sufficient

p01.l30.: "these Himalayan glacier" > "Glaciers in the HKH"

p02.l51.: "The reported glacier change" > "The reported glacier area change"

p02.l55.: "information" > "knowledge"

p02.l55.: "no study exist on the ... and ... and .. and ...". True, but individual aspects are analyzed. Please change.

p02.l59.: "during studied period" > "during the studied period"

p02.l61.: "also evaluated" $\sim$ (meaning strange wording)

p02.l63.: "done by Das&Shrama 2019"

p02.l64.: How special are these points?

p02.l69.: "of mighty Chenab" > "of the mighty Chenab", maybe another word for mighty

p03.l75.: "highly active glaciers", what is meant here?

p03.l80.: it took me a while to figure out what this figure (a) was doing here. does this really contribute here to the work? and does it give insights?

p04.: This can be shortened considerably. The data is similar to the one used in Das&Shrama'19, one line of text might suffice. You read a lot about SRTM, while a simple bias is used from the literature.

p05.l123.: "huge ice cliff", please give something in a metric unit

p05.l124.: "thick layer", what is thick and is it homogenous?

p05.l127.: "recorded", what do the authors mean "surveying/observing"?

p07.l146.: "considered as are" $\sim$

p07.l147.: "all criterion was applied" > "were"

p07.l152.: why not use the "box"-method of (Moon & Joughin)?

p08.l173.: "penetrates more" into the snowpack

p08.p4.3.1.: reduce to one sentence

p08.p4.3.2.: can be simplified

p10.l217.: please include reference, do the authors mean the procedure of (Rolstad et al.)?

p10.l229.: Cosicorr is not normalized cross correlation

p10.l235.: it is not nadir looking, it has a swath, it is a pushbroom

p11.l242.: isn't there saturation in the green band of ETM+ 8bit data, why not use NIR, which is typically done with older data?

p11.l246.: please say why you do this

p11.l250-259.: this reads like a text from a manual, please reduce.

p14.l308.: "select glaciers"

p17.l343.: "glacier in the JCW" > +s

p17.l343.: "is active"... what is meant here?

p17.l344.: "for analyzed" please change

p17.l350.: what is meant here, I don't get this out of the table. And what does this mean?

p17.l352.: "of select nine", please change

p20.l388.: "the first", GOLIVE exist for some years, see (Fahnestock et al.)

p20.l392.: "in ablation and accumulation zone" wording

p20.l392.: "that highly"

p20.l393.: "while accumulation zone"

p20.l396.: "range broadly"

p20.l403.: "with a reasonable request", what do the authors mean?

p21.l408.: "the regional climate glacier interactions", what is meant here?

---

## Referee Comment (RC2) · Anonymous Referee #2 · 4 Mar 2020

This manuscript presented the glacier terminus retreat, mass budget, and surface velocity measurements for the Jankar Chhu Watershed, Lahaul Himalaya, India using remote sensing and field observations. Such an investigation is definitely welcome to understand the impact of current climate warming on Himalayan glaciers. However, there are many issues with the overall structure, scientific language, and presentation of the work particularly with the repetition of work that will not create any significant contribution in glacier research. The language and grammar of the manuscript also need a thorough revision. Major Comments. Objective 1: The first objective is completely repetition of previous work, which does not contribute anything new for the research community. As most of the results related to glacier change already

published in another journal by the same authors (i.e. Das & Sharma 2019; Journal of Glaciology). In addition, taking consideration of nine studied glaciers will not fulfil the objective of ESSD journal to provide high-quality regional data to Earth System Sciences. Objective 2: Several studies have provided better data/information on surface elevation and geodetic mass balance for the study region (Brun et al. 2017; Maurer et al. 2019; King et al. 2019). Some studies have reported the long-term surface elevation change for the past four decades (King et al. 2019; Maurer et al. 2019). Vijay & Braun (2016) also analysed the geodetic glacier mass balance change in the Lahaul Himalaya from 2000 to 2012 then just for two more years mass balance change computed in the present study (till 2014) did not show any novelty. Also, Vijay & Braun (2016) have provided their data in the public domain and covered the large region. The geodetic and velocity changes for glaciers of Lahaul Himalaya already available freely from NSIDC (https://nsidc.org/data/HMA_GlacierAvg_dH). Therefore, the present study does not significantly contribute to the current knowledge of the status of glaciers in Western Himalaya. Objective 3: Recently a number of studies produced or developed velocity data for Himalayan glaciers and some data are freely available from NSIDC (https://nsidc.org/data/golive). I would suggest to compute seasonal variations in velocity instead of annual changes (e.g. Scherler et al. 2008; Satyabala 2016) and connect with mass balance changes which are missing in this study. Specific Comments: Abstract: Page 1 Line 20: "Field observations/measurements also support the findings". I could not find comprehensive ground data that support the validation of velocity and mass balance changes. Most of the fieldwork shows some photographs of glacier features and debris cover thickness. Such limited field data just provide the glimpse but does not validate the results. Introduction: Page 2 Line 41: It may be ELA elevation. Page 2 Line 41: You may include surface elevation change or geodetic mass balance as same calculated in the presents study. Page 2 Line 48-50: It would be better to revise the sentence as in recent years' number of studies used High resolution declassified images for this part of the region (Negi et al. 2013; Chand & Sharma 2015a; Chand & Sharma 2015b; Chand & Sharma 2016; Chand et al. 2017). Page 2 Line 51:

There is a need to provide gaps area for the Lahaul region instead of directly jumps into the comparison of retreat rate for the glaciated basin of Lahaul Himalaya. Page 2 Line 55: Several studies carried out a detailed analysis of geodetic mass balance for the Lahaul Himalaya including study area (Vijay & Braun 2016; Brun et al. 2017; Maurer et al. 2019; King et al. 2019). Some studies extended glacier surface elevation change or mass balance change for the past three decades to 2016-2017 (Maurer et al. 2019; King et al. 2019). Therefore, in this case, it is not true that "no study exists for this region". Data Sources & Methodology: Page 4 Line 104-106: There is a need to provide the brief description about the correction of elevation difference due to change in vertical reference between these two used DEM datasets (Mukherjee et al. 2013). In addition, you may use recently available high-resolution DEMs for Himalaya e.g. HMA DEM (8 m) or ALOS DEM (12.5 m) for mass balance estimation. Page 4 Line 114: Justification is required to use ASTER DEM as this study already used two DEMs. Page 5 Line 122: You may use the name of the glacier, as it has been written in Survey of India topographical map. Page 5 Line 126: There is a need to provide the uncertainty of Infrared (IR) Laser Thermometer sensor. What is the use of this data? How you will link single time observation with mass change and surface velocity estimation, which is the primary objective of the study? Page 5 Line 130-131: The handheld GPS measurements are collected for a few points, which can be used in general but does not contribute much in mass change and surface velocity estimation. Page 9 Line 188: Why select 25 degrees threshold where most of accumulation or ablation part has such slope. Justification is needed to use this slope threshold whereas most of the glaciological community use threshold of 40-45 degrees. Page 9 Line 202: Why used 2016-glacier boundary instead of 2014 as you used the 2014 TANDEM for the geodetic mass balance estimation?

References Brun F, Berthier E, Wagnon P, Kääb A, Treichler D. 2017. A spatially resolved estimate of High Mountain Asia glacier mass balances from 2000 to 2016. Nat Geosci. 10(9):668–673. Chand P, Sharma MC. 2015a. Frontal changes in the Manimahesh and Tal Glaciers in the Ravi basin, Himachal Pradesh, northwestern Himalaya (India), between 1971 and 2013. Int J Remote Sens [Internet]. [accessed 2015 Aug 27] 36(16):4095–4113. http://www.tandfonline.com/doi/abs/10.1080/01431161.2015.1074300?journalCode=tres20#.Vd7ggPaqqko Chand P, Sharma MC. 2015b. Glacier changes in the Ravi basin, north-western Himalaya (India) during the last four decades (1971–2010/13). Glob Planet Change [Internet]. [accessed 2015 Oct 28] 135:133–147. http://www.sciencedirect.com/science/article/pii/S0921818115300953 Chand P, Sharma MC. 2016. Monitoring Frontal Changes of Shah Glacier in the Ravi Basin, Himachal Himalaya (India) from 1965 to 2013. Natl Acad Sci Lett. 39(2):109–114. Chand P, Sharma MC, Bhambri R, Sangewar C V, Juyal N. 2017. Reconstructing the pattern of the Bara Shigri Glacier fluctuation since the end of the Little Ice Age, Chandra valley, north-western Himalaya. Prog Phys Geogr [Internet]. [accessed 2017 Oct 25] 41(5):643–675. http://journals.sagepub.com/doi/10.1177/0309133317728017 Das S, Sharma MC. 2019. Glacier changes between 1971 and 2016 in the Jankar Chhu Watershed, Lahaul Himalaya, India. J Glaciol. 65(249):13–28. King O, Bhattacharya A, Bhambri R and Bolch T 2019. Glacial lakes exacerbate Himalayan glacier mass loss. Scientific Reports, 9(1), pp.1-9. Maurer JM, Schaefer JM, Rupper S, Corley A. 2019. Acceleration of ice loss across the Himalayas over the past 40 years. Sci Adv. 5(6). Mukherjee Sandip, Joshi PK, Mukherjee Samadrita, Ghosh A, Garg RD, Mukhopadhyay A. 2013. Evaluation of vertical accuracy of open source Digital Elevation Model (DEM). Int J Appl Earth Obs Geoinf [Internet]. [accessed 2014 May 27] 21:205–217. http://linkinghub.elsevier.com/retrieve/pii/S030324341200195X Negi HS, Saravana G, Rout R, Snehmani. 2013. Monitoring of great Himalayan glaciers in Patsio region, India using remote sensing and climatic observations. Curr Sci. 105(10):1383–1392. Satyabala SP. 2016. Spatiotemporal variations in surface velocity of the Gangotri glacier, Garhwal Himalaya, India: Study using synthetic aperture radar data. Remote Sens Environ. 181:151–161. Scherler D, Leprince S, Strecker M. 2008. Glacier-surface velocities in alpine terrain from optical satellite imagery—Accuracy improvement and quality assessment. Remote Sens Environ [Internet]. [accessed 2013 Jun 4]
112(10):3806–3819.     http://linkinghub.elsevier.com/retrieve/pii/S0034425708001934
Vijay S, Braun M. 2016.    Elevation change rates of glaciers in the Lahaul-Spiti
(Western Himalaya, India) during 2000-2012 and 2012-2013. Remote Sens [Internet].
[accessed 2017 Feb 5] 8(12):1038. http://www.mdpi.com/2072-4292/8/12/1038

---

## Editor Comment (EC1) · Reinhard Drews (Editor) · 7 May 2020

After considering the reviews and editor recommendation, the authors have decided not to submit a revised version of the paper.